**www.cambridge.org/ext**

## Perspective

Holocene; Pleistocene; Anthropocene; biodiversity; rewilding

**Corresponding author:**
Barry W. Brook;
Email: barry.brook@utas.edu.au

# Uncooling the planet: Rewilding for function in a post-Pleistocene climate

Barry W. Brook[1] and Guy F. Midgley[2]

[1]School of Natural Sciences, University of Tasmania, Sandy Bay, TAS, Australia and [2]Department of Botany and Zoology, Stellenbosch University, Stellenbosch, Western Cape, South Africa

## Abstract

The cold, low carbon dioxide ($CO_2$) conditions of the Pleistocene epoch fundamentally structured ecosystems, profoundly influencing the evolutionary trajectory of *Homo sapiens* and other large mammals. Although often considered uniquely stable, the Holocene is more usefully viewed as just another Pleistocene interglacial interval that was naturally trending towards a renewed glacial phase. However, rapid anthropogenic greenhouse gas emission rates have reversed this trajectory and might have now foreclosed the prospect of returning to cyclic glacial climates for millennia. A large set of flora and fauna has benefited from low $CO_2$ conditions, which we define as low-$CO_2$ dependents. By elevating atmospheric $CO_2$ concentrations beyond levels seen for millions of years, we have accelerated global warming beyond the adaptive capacities of many species and ecosystems. African savannas and grasslands are particularly relevant in this context because this was the environment in which the human species evolved. These biomes have been previously maintained by fire and carbon scarcity but are now experiencing woody encroachment driven by rising $CO_2$. The resultant global reforestation further threatens biodiversity adapted to open ecosystems, while rewilding initiatives must therefore pair prehistoric analogues with explicit climate-fitness tests that anticipate mid-century $CO_2$ trajectories. Addressing these complex challenges requires both targeted local interventions and systemic policy reforms, grounded in a pragmatic recognition of the transient nature of the Holocene. Recognising the transience of any single baseline allows conservation and agriculture to plan for a dynamic, overshoot-prone future.

## Impact statement

This research reframes contemporary climate debates by highlighting the fundamental shift in planetary ecological conditions represented by overshooting Paris Accord targets. Placing the Holocene/Anthropocene debate within a longer Pleistocene context of glacial–interglacial cycles frames the failure to meet Paris Accord targets more properly within a two-million-year cycle of cool, low-$CO_2$ conditions. Recognising this deeper timescale could better inform policy decisions and guide climate-integrated conservation strategies to protect low-$CO_2$-dependent ecosystems. High projected extinction rates in biodiversity hotspots are a clear warning flag for low-$CO_2$-dependent ecosystems. Equally concerning is the rapid pace of observed ecological change, notably woody encroachment, now unfolding in subtropical savannas. These perspectives are relevant when considering global rewilding, de-extinction and anticipatory ex-situ conservation discussions that have immediate investment implications and deeper intergenerational equity consequences.

## Introduction

The Holocene is routinely treated as the planetary 'normal'. In a stratigraphic context, however, it is just the most recent interglacial pulse within the 2.6-million-year Pleistocene glacial–interglacial oscillator (Steffen et al., 2018; Hobart et al., 2023). Its climate signal is neither warmer nor longer-lived than that of earlier interglacials, such as Marine Isotope Stage 5e (~125 thousand years [kyr] ago). What is unique is that *Homo sapiens* civilisations matured during this lull. Indeed, van der Pluijm (2014) has argued that the 'golden spike' accepted by the International Commission on Stratigraphy (http://goo.gl/GFdeit) represents a '*moderate atmospheric signal [that] characterizes the Pleistocene-Holocene boundary as currently defined, with the end of the latest [...] glacial as its primary geologic signature (yet, the process of receding ice is similar among all Quaternary interglacials)*'. Furthermore, pre-industrial Holocene warmth was milder than the previous interglacial – referenced as a benchmark of excessive anthropogenic interference in the planetary climate (Hansen et al., 2012).

Contemporary discussion of anthropogenic impacts on fauna and flora often overlooks this deeper temporal lens. This blind-spot obscures an ecologically and societally relevant perspective on human disturbance of the quasi-periodic rhythm of glacial–interglacial cycles (Figure 1) that

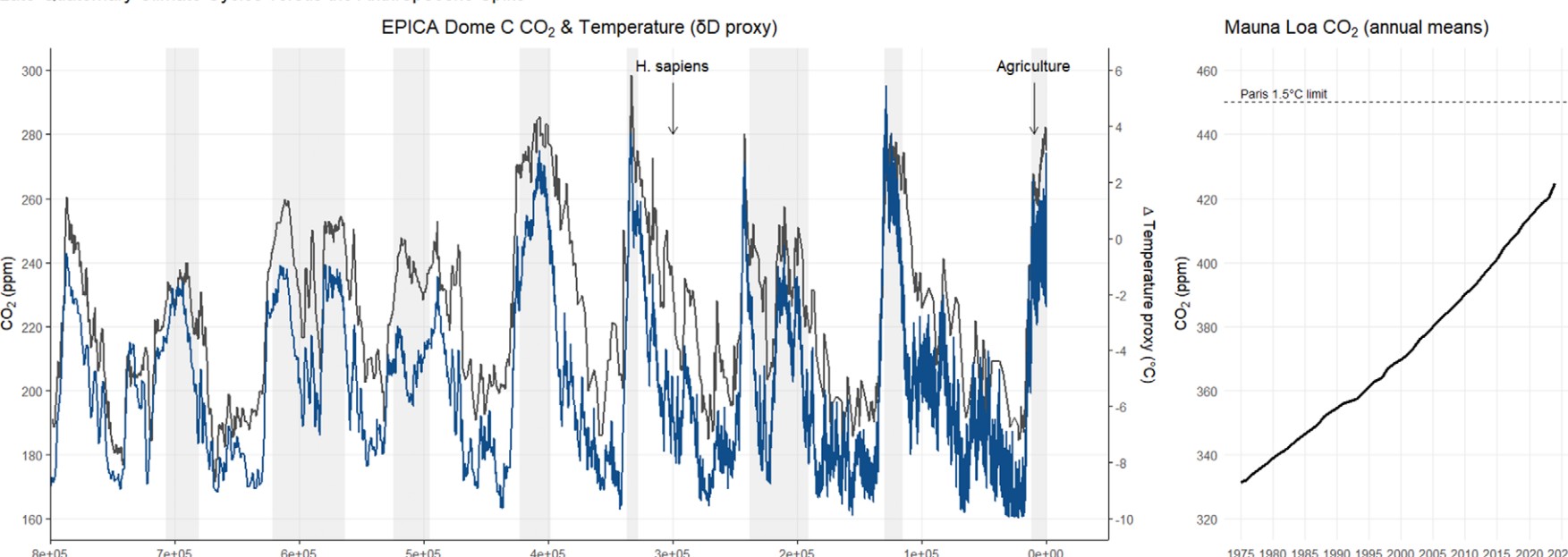

**Figure 1.** Late-Quaternary climate cycles and the modern $CO_2$ spike. Left panel (0–800 ka BP, reversed x-axis): Black line: atmospheric $CO_2$ (ppm) from the EPICA Dome C ice-core composite. Blue line: Antarctic temperature anomaly ($\delta$D proxy) rescaled to the $CO_2$ axis ($-10\,°C \to 160$ ppm; $+6\,°C \to 300$ ppm). Grey bands: recognised warm Pleistocene interglacials over this period (marine isotope stages 1, 5e, 7c–a, 9e, 11c, 13a, 15a, 17c). Vertical arrows: estimated emergence of *Homo sapiens* (~300 ka), first widespread agriculture (~10 ka), and the current rapid overshoot of Pleistocene $CO_2$ maxima (~20th century onward). Right panel (1975–2024 CE): Black line: Mauna Loa annual mean $CO_2$. Dashed line: 450 ppm, the upper bound compatible with a >66 % chance of holding warming to 1.5 °C (IPCC, 2021). Data sources: EPICA $CO_2$ and $\delta$D series: NOAA NCEI Paleoclimatology archive (www.ncei.noaa.gov). Mauna Loa $CO_2$: NOAA GML (Keeling curve).

Barry W. Brook and Guy F. Midgley

has shaped Earth's recent geological history (Lüthi et al., 2008). Today, rapid anthropogenic warming has displaced Earth's climate trajectory from not only the Holocene baseline (Hansen et al., 2012) but the broader Pleistocene regime (Jackson and Blois, 2015), necessitating a forward- rather than backwards-looking framework for conservation and rewilding. Here we argue for a reframing of the debates on anthropogenic influence by emphasising that the Anthropocene Epoch itself may herald an incipient climate and carbon cycle transition well beyond Pleistocene conditions.

This transition is unparalleled in its speed and magnitude, with atmospheric carbon dioxide ($CO_2$) concentrations already exceeding 420 ppm: levels significantly higher than those of any natural fluctuation recorded during past glacial–interglacial cycles (Mann et al., 2019). The consequences for ecosystems are already profound, especially for species and habitats adapted to arid, carbon-limited environments (Barnosky et al., 2012; Urban, 2015). As these shifts intensify, conventional assumptions guiding conservation, rewilding and agriculture, premised on Holocene stability, must be re-examined, particularly from the standpoint of species whose evolutionary legacies were forged largely within the Pleistocene's dynamic conditions.

### Deep-time context: The Pleistocene and its cycles

The Pleistocene was defined by repeated oscillations between cold glacial periods and brief, warmer interglacials, with cold, low-$CO_2$ phases dominating the ecological context for most of the late Quaternary, favouring grassland expansion and megafaunal proliferation (Mann et al., 2019). During glacials, global temperatures fell by several degrees Celsius relative to present conditions, and atmospheric $CO_2$ dropped as low as 180 ppm (Lüthi et al., 2008; Tripati et al., 2009). Massive ice sheets expanded over northern latitudes, sea levels fell dramatically and arid conditions spread across many regions. In this carbon-scarce world, plants using the $C_3$ photosynthetic pathway would experience reduced quantum yield (Ehleringer, 2005), with woody plants particularly severely affected due to their relatively higher demand for structural carbon (Bond and Midgley, 2012), especially when compared to $C_4$ grasses (Ward et al., 2008). Open biomes, such as grasslands, shrublands and savannas, proliferated globally (Malhi et al., 2016). This cyclical alternation, with a cooler, low-$CO_2$ bias, rather than persistent warmth, represents the planet's ecological baseline throughout the Pleistocene.

African ecosystems, including savannas and grasslands, likely expanded once atmospheric $CO_2$ dropped below about 500 ppm (but see Osborne, 2007), likely amplified by increasing wildfire prevalence (e.g., Miao et al., 2022), and key succulent plant groups in arid biomes like the succulent Karoo evolved explicitly under cold, low-$CO_2$ conditions (Klak et al., 2004). Sparse tree growth, frequent fires and herbivore-driven vegetation control maintained expansive, open landscapes, allowing diverse communities of large herbivores, such as wildebeest, elephants and zebras, to flourish (McKee, 2001). Africa's iconic mammal assemblages and bird communities co-evolved with this interplay between limited woody vegetation, abundant grasses and recurring drought (Huntley et al., 2016), a regime maintained by large-herbivore impacts and wildfire feedbacks (Malhi et al., 2016). These ecosystems persisted resiliently across numerous glacial cycles, their ecological structure continually shaped by the constraints imposed by low atmospheric $CO_2$. Late Cenozoic faunal composition and ecosystem structure in Africa were far from static, with repeated turnover in large-herbivore guilds and biome shifts across the late Cenozoic and Late Pleistocene (Faith et al., 2024), and it appears that the African bovid browsers drove the evolution of thorniness and spininess in African savannas before the establishment of fire-dependent savannas (Charles-Dominique et al., 2016).

Early *H. sapiens* emerged within this challenging ice-age context, with small, mobile groups surviving primarily through hunting, scavenging and foraging (Richerson et al., 2001). These scattered human populations likely endured repeated evolutionary bottlenecks during particularly severe glacial intervals, periods marked by scarcity in reliable plant resources and intense climatic instability (Timbrell, 2024). This adversity favoured dietary versatility and behavioural adaptability. Over successive generations, humans honed sophisticated toolkits, cooperative social networks and technological innovations to endure harsh environmental pressures, and had major impacts on regional fauna (Cooper et al., 2015). Ultimately, these hard-won adaptations laid essential foundations for a global human expansion once climates warmed, resources diversified and conditions became more stable.

### $CO_2$ starvation, the agricultural switch and human diet

During glacial maxima, low atmospheric $CO_2$ levels severely restricted plant growth, keeping wild cereals and other $C_3$ species near the lower edge of their photosynthetic capability (Ward et al., 1999). With concentrations near 180–200 ppm, the productivity of many agriculturally relevant $C_3$ plants was depressed and yields unreliable; experiments on wild crop progenitors and model $C_3$ taxa consistently show strong $CO_2$ limitation in this range. Under such conditions, cultivation offered little advantage over foraging. It is noteworthy that this does not imply ecosystem-wide primary-productivity collapse ($C_4$-dominated steppe–savanna primary production could still support diverse megafauna), but it does explain why early farming returns were poor (Cunniff et al., 2017). Early human populations therefore remained small, mobile and reliant principally upon foraging and hunting, limiting their potential to establish larger or more permanent settlements. Moreover, the widespread extinction and decline of large mammals during the Late Pleistocene further constrained reliable access to animal-based food resources, possibly increasing human incentives to explore plant cultivation as an alternative nutritional strategy (Ben-Dor and Barkai, 2024).

Following the last glacial maximum, atmospheric $CO_2$ gradually climbed from roughly 200 to about 270 ppm, significantly boosting photosynthetic efficiency for many crops (Ward et al., 2008). Experimental evidence suggests yields of cereals like wheat and barley could have increased substantially with $CO_2$ elevations in this critical range (Cunniff et al., 2017). The post-glacial $CO_2$ rise thus removed a key physiological lid on $C_3$ photosynthesis (Ward et al., 2008), fundamentally changing the human ecological landscape. For the first time, systematic farming became viable, supporting stable communities, population growth and the creation of food surpluses. The independent emergence of agriculture across several regions worldwide aligns closely with this period of rising $CO_2$, suggesting that $CO_2$ was a primary environmental constraint on agricultural productivity rather than merely a background climatic factor (Richerson et al., 2001). Thus, rising $CO_2$ likely interacted synergistically with declining megafaunal availability, together creating conditions favourable to the independent emergence and global spread of agriculture.

Modern agricultural systems have their roots in this post-glacial increase in atmospheric $CO_2$. Yet, today's $CO_2$ concentrations now exceed 420 ppm, far surpassing anything experienced during the

entire Pleistocene epoch (Lüthi et al., 2008). Although elevated $CO_2$ can enhance crop productivity and improve water-use efficiency, these potential benefits are increasingly overshadowed by climate instability. Rising temperatures, altered precipitation patterns and more frequent extreme weather events threaten agricultural stability worldwide (Barnosky et al., 2012). Humanity's food systems now rely precariously on a balance between the fertilising effect of higher $CO_2$ and the disruptive impacts of rapid climate change. This balance is becoming ever harder to maintain in the face of ongoing anthropogenic warming.

### Why the Holocene is (mostly) just another interglacial

The Holocene epoch, beginning ~11,700 years ago, often appears uniquely significant because it encompasses the rise of agriculture and complex human civilisations. Yet, when viewed through a broader geological lens, it closely resembles earlier interglacials (such as the Eemian around 125,000 years ago) when atmospheric $CO_2$ concentrations and temperatures also exceeded typical glacial conditions (Lüthi et al., 2008). Such intervals routinely punctuated the otherwise dominant cold phases of the Pleistocene, typically lasting 5–15 kyr, before conditions reverted to glacial climates (Figure 1). The perceived stability of the Holocene stands out primarily because humans flourished within it, not because its climatic profile differs fundamentally from previous interglacials. Our human-centred perspective thus risks obscuring the geological reality: under normal orbital cycles, the Holocene, too, would eventually have faded into another glacial episode (Hobart et al., 2023).

Despite its centrality to human history, under natural orbital forcing and absent anthropogenic greenhouse gases, the present interglacial was likely to persist for tens of millennia – on the order of ~50 kyr – given the current weak-eccentricity configuration (Berger and Loutre, 2002). Modest anthropogenic $CO_2$ loads now appear sufficient to postpone glacial inception for ≥50–100 kyr. By most measures (temperature trends, rising $CO_2$ levels and sea-level increases), the Holocene fits comfortably into established patterns that characterise past interglacial peaks (Marcott et al., 2013). Its apparent uniqueness is circumstantial: complex societies probably flourished because the interval was unusually stable, not because it broke the mould of prior interglacials (Marcott et al., 2013). By adopting the Holocene as our implicit baseline for 'normal' conditions, we risk significantly underestimating Earth's inherent climatic variability, as well as mischaracterising the profound and unprecedented consequences of current anthropogenic warming. Recognising this broader deep-time context is critical if we aim to make informed decisions about conservation, agriculture and climate policies in an increasingly uncertain future.

### Uncooling the planet: The Anthropocene shift

The unprecedented rise in atmospheric $CO_2$ has resulted primarily from fossil fuel combustion and land-use changes, with greenhouse gas emissions accelerating exponentially over the past two centuries (Hansen et al., 2012). Previously, during natural deglaciations, atmospheric $CO_2$ typically rose <100 ppm over millennia. In 170 years, we added ~140 ppm $CO_2$: roughly the full glacial–interglacial swing, but 40–100 times faster. Orbital forcing no longer governs global climate; anthropogenic combustion of fossil fuels now drives rapid ecological and climatic transformations that outpace evolutionary and ecological adaptation (Steffen et al., 2018).

The extraordinary pace of this transformation presents an exceptional ecological challenge, particularly for species and ecosystems evolved to cope with slower, orbitally driven climate fluctuations (Tripati et al., 2009). Many organisms relied on gradual migration or incremental genetic adaptation to match shifting temperatures and environments. Pleistocene biota did experience abrupt swings such as Dansgaard–Oeschger and Heinrich events (decadal–centennial onsets), with 8–16 °C jumps recorded over Greenland; however, these were likely regionally focused and transient within glacial backgrounds, unlike today's globally coherent, $CO_2$-driven, multi-century warming (IPCC, 2021). Now, with climate conditions changing within human lifetimes, ecosystems face a markedly increased risk of local or regional collapse, and potentially, global extinctions (Barnosky et al., 2012). We are rapidly leaving behind the climatic conditions under which many extant species evolved, pushing Earth into an ecological regime that many living populations have never experienced before.

Further compounding this challenge are overshoot scenarios. These are situations in which global temperatures temporarily rise significantly above moderate warming targets before possibly stabilising at lower levels (Steffen et al., 2018). Overshoot may force ecosystems past biogeochemical tipping points, for example, coral thermal limits and savanna grass–tree hysteresis thresholds around ~500 ppm $CO_2$, with little prospect of reversal on human timescales even after temperatures peak (Barnosky et al., 2012). Even if subsequent cooling is achieved, the damage to habitats and species may prove permanent. This possibility significantly raises the ethical and practical urgency of mitigating climate change promptly, as a return to previously stable conditions may prove neither feasible nor guaranteed.

### Case study: Implications for African biodiversity

African ecosystems like the expansive savannas and grasslands and distinctive succulent habitats were the environments in which modern humans evolved (Timbrell, 2024). African biota largely flourished under post-Miocene conditions of low atmospheric $CO_2$, pronounced seasonality, frequent fire and harsh droughts (Cohen et al., 2007). Today, rapidly rising $CO_2$ is altering this ecological balance, favouring woody plants that grow more vigorously and better withstand fire. This process, termed woody encroachment, transforms open habitats into denser woodlands, reducing grassland area and severely impacting large herbivores and their predators (Parr et al., 2014). Because woody encroachment suppresses landscape flammability and grazer carrying capacity, such shifts can outpace herbivore or predator adaptation, magnifying extinction risk.

Ironically, well-intentioned climate interventions can exacerbate this ecological transformation. Global tree-planting initiatives frequently target open landscapes for afforestation, incorrectly presuming these naturally treeless habitats are degraded (Kumar et al., 2020), thereby extinguishing the herbivore-maintained light regime on which $C_4$ grasses depend. Introducing large numbers of trees into grassland and savanna ecosystems undermines specialist species and disrupts vital ecological processes unique to open habitats (Zaloumis and Bond, 2011). Avoiding these unintended consequences demands a deeper, regionally granular understanding of healthy grassland and savanna systems, rather than default assumptions about barren landscapes needing trees.

Many iconic African species depend heavily on open, fire-maintained ecosystems: precisely the habitats threatened by rising $CO_2$ levels and increased woody growth (Western et al., 2021).

Succulent plants adapted to arid conditions, grazing mammals dependent on open, fire-maintained landscapes, predators specialised on these grazers and grassland-specialist birds, all face considerable risk if tree encroachment advances unchecked (Stevens et al., 2017). These 'low-$CO_2$-dependent' species possess adaptations finely tuned to low-canopy environments, such as specialised water conservation strategies or cooperative grazing behaviours. As landscapes transition to dense woodland, these adaptations may turn from strengths into vulnerabilities, placing entire evolutionary lineages in jeopardy.

Yet, despite clear threats, global ecological assessments often simplify habitat changes into broadly positive narratives of 'greening', masking severe local vulnerabilities (Parr et al., 2014). Models highlighting net vegetation gains frequently overlook critical losses in grassland-specialist biodiversity and functional ecological processes, such as herbivore-mediated nutrient cycling and seed dispersal (Perino et al., 2019). This omission becomes especially critical under overshoot scenarios, where ecosystems could degrade irreversibly before meaningful intervention occurs. Therefore, more sophisticated analyses that are sensitive to the unique dynamics of open ecosystems and their vulnerability to rapid carbon-driven shifts and land-use change are urgently needed to guide effective conservation strategies (Andersen et al., 2005).

### From analogy to prognosis: Why functional proxies must pass a future-fitness screen

Here, we develop a screening framework primarily for terrestrial megafaunal mammals used as functional proxies in rewilding, that is, extant taxa introduced to deliver specified ecological functions (e.g., grazing, browsing, seed dispersal and carrion provisioning) of extinct or locally extirpated counterparts, irrespective of phylogenetic fidelity. The general logic extends to other vertebrate groups, but metrics would need recalibration.

Past climate analogues provide useful benchmarks only if the desired ecological states remain climatically attainable under foreseeable trajectories. CMIP6 ensemble means project global mean temperatures of +2.7 to +3.6 °C and $CO_2$ of ~560–700 ppm under SSP3–7.0 and SSP5–8.5 by 2100 (IPCC, 2021). At those concentrations, leaf-level $C_3/C_4$ competitiveness, fire-return intervals and soil-moisture regimes diverge sharply from Pleistocene patterns that shaped megafaunal guilds. In this context, any candidate 'proxy species' must clear these two climatic (heat and $CO_2$-related) filters.

To operationalise the filters for African systems under mid-century SSP3–7.0/SSP5–8.5 conditions, we distil four trait axes with simple, field-usable tests:

i) *Heat tolerance.* Prioritise species whose thermoregulatory and evaporative-cooling capacity permits routine activity under high heat loads; as a rule-of-thumb, exclude taxa failing at upper critical wet-bulb ~30–32 °C or with narrow thermoneutral zones. The large-mammal heat stress literature supports these bounds (Fuller et al., 2016). Rationale: Lethal/prolonged sub-lethal heat undermines foraging time and function delivery.

ii) *Diet breadth and fermentation.* Prefer mixed feeders and hind-gut fermenters that maintain faecal nitrogen content at or above 1.5% under woody-biased diets (Leslie and Starkey, 1985); this threshold is a widely used adequacy marker in cervids and works heuristically across herbivores (to be locally validated). Rationale: $CO_2$-driven woody encroachment shifts forage quality/availability.

iii) *Thermal-niche plasticity.* Favour species with broad thermal performance (e.g., demonstrably wide thermoneutral zone or low heat-sensitivity of maintenance metabolism) and behavioural flexibility (shade use and nocturnality) (Speakman et al., 2004; Mitchell et al., 2018). Rationale: Lowers management inputs under warming.

iv) *Trophic redundancy.* Prioritise functions that can be delivered by ≥2 congeners or functional analogues within the region. Rationale: Reduces single-point failure risk under climatic shocks; supports continuity of function.

*Cross-cutting disturbance compatibility.* Because fire–grazer feedbacks act to structure savannas, candidate proxies should tolerate and, ideally, help maintain targeted fire regimes (pyric-herbivory). This is not a fifth criterion so much as a constraint on deploying the four axes.

Collectively, these thresholds screen out cold-adapted specialists, such as musk ox (*Ovibos moschatus*), and highlight heat-tolerant generalists, for example, African buffalo (*Syncerus caffer*) and plains zebra (*Equus quagga*) as viable proxies for mid-century SSP3–7.0 climate conditions (IPCC, 2021), provided they are deployed within fire regimes that sustain open structure.

### Rewilding in a dual-baseline framework

Modern rewilding seeks to restore key trophic processes, often by re-introducing megafauna that dominated Pleistocene food webs (Donlan et al., 2006). Yet, long-term ecological success hinges explicitly on whether those proxies retain their functional roles under elevated atmospheric $CO_2$ levels, expected to surpass ~550 ppm by the late twenty-first century under SSP3–7.0 (IPCC, 2021). Ambitious efforts, such as 'Pleistocene Park' in northern Siberia (Popov, 2020), seek to recreate expansive grasslands formerly sustained by mammoths and other megafauna, involving both rewilding and, potentially, de-extinction (Novak, 2018). Deep-time analogues, although instructive for initial candidate selection, risk becoming maladaptive unless taxa pass explicit climate-fitness screens based on heat tolerance, diet flexibility, thermal plasticity and functional redundancy, as defined in the previous section ('From Analogy to Prognosis'). As argued herein, today's global climate is rapidly diverging from the cool, low-$CO_2$ conditions that originally shaped these landscapes. Even if certain extinct or diminished species are revived through selective breeding or genetic engineering, the environmental context in which they evolved no longer exists. This fundamental ecological mismatch raises questions about the feasibility of sustaining such reintroduced populations amid rapidly warming climates and changing vegetation regimes, but also about whether, and to what degree, such fauna revitalisation might act to restore lost ecosystem function (Lundgren et al., 2020).

To remain viable, rewilding strategies must be forward-looking rather than simply recreating past conditions. Reintroducing large herbivores or keystone predators can indeed help sustain open ecosystems, but only if these interventions explicitly consider ongoing climatic changes and contemporary land-use pressures (Seddon et al., 2014). Conservationists may increasingly need to embrace novel assemblages that prioritise ecological functionality over strict evolutionary accuracy. Successful rewilding projects will depend critically on adaptive management practices explicitly aligned with future climatic conditions, such as targeted selection of taxa that fulfil necessary ecological roles under anticipated $CO_2$ and temperature regimes.

Given these considerations, rewilding is best conceptualised as an adaptive ecological experiment, guided by three explicit questions addressing trophic functions, tipping points and management interventions:

### Question 1: Which trophic functions are climate-agnostic?

Grazing-induced seedling suppression, carcass-linked nutrient hotspots and long-range seed dispersal can persist across wide thermal envelopes. For example, evidence from Yakutia's Pleistocene Park demonstrates that even modest densities of bison (*Bison bonasus*), horses (*Equus ferus*) and reindeer (*Rangifer tarandus*) (~5, 7.5 and 15 individuals km$^{-2}$, respectively) maintain grass-forb dominated landscapes despite a regional warming of ~3 °C (Zimov et al., 2012). Ultimately, effective rewilding must balance fidelity with foresight, selecting species whose traits remain functional under projected temperature, $CO_2$ and disturbance regimes (Perino et al., 2019), so that evolutionary function is preserved without forfeiting climate-fitness under foreseeable overshoot trajectories.

### Question 2: At what $CO_2$-temperature intersections do those functions fail?

Process-based vegetation models (e.g., Lund-Potsdam-Jena General Ecosystem Simulator: LPJ-GUESS or adaptive Dynamic Global Vegetation Model: aDGVM) converge on a non-linear response: once atmospheric $CO_2$ exceeds roughly 480–530 ppm, $C_3$ saplings gain a photosynthetic and water-use advantage that even doubled grazer biomass cannot fully offset (Bond and Midgley, 2012; Higgins and Scheiter, 2012). Below ~500 ppm $CO_2$, moderate grazing and episodic fire effectively maintain savanna openness. Within the 480–530 ppm 'tipping band' identified by vegetation models (Higgins and Scheiter, 2012), woody cover accelerates non-linearly. Above it, landscapes tend towards persistent woody states unless grazer or browser pressure intensifies two- to fourfold relative to pre-industrial baselines (O'Connor et al., 2014). As such, pragmatic management would either strive to keep regional $CO_2$ levels beneath this critical tipping band via aggressive mitigation, or else proactively relocate vulnerable ecosystem functions into edaphically fire-limited regions, where the threshold for woody encroachment is inherently higher (Archibald et al., 2019).

### Question 3: What management buffers exist?

Practical interventions, including explicit fire-regime design (patchy, seasonally timed pyric-herbivory), $CO_2$-sensitive thinning of sapling cohorts and dynamic stocking density adjustments, can together enhance open-ecosystem resilience, potentially accommodating an additional 0.6–1.1 °C of warming before critical tipping points occur (Barnosky et al., 2012; Lundgren et al., 2020; Western et al., 2021). Crucially, such strategies should be formalised within adaptive management frameworks well before early signals of ecosystem collapse become evident, ensuring timely and effective implementation.

### Broader ethical and policy reflections

Preserving species and habitats that evolved under glacial conditions presents significant ethical challenges, notably regarding how far societies should go to protect these 'ice-age legacies' (Donlan et al., 2006). Rapid anthropogenic warming increasingly threatens their survival in situ, raising difficult questions about the role of ex situ conservation, assisted migration or other proactive interventions. Such strategic triage depends on explicit value judgements (e.g., do we privilege deep-time lineages, functional redundancy or climate-robust proxies?) and on transparent evaluation of future fitness. In practice, proactive conservation might necessitate active and potentially controversial ecological management, such as routinely removing encroaching woody plants to maintain open savanna ecosystems. Yet, these actions blur the distinction between traditional conservation and deliberate ecological engineering, prompting deeper reflections on our ethical responsibilities not only to protect evolutionary legacies but to proactively engineer ecosystems capable of thriving in rapidly shifting climatic contexts (Perino et al., 2019).

Systemic economic barriers compound these ethical considerations. Economic incentives frequently prioritise short-term outcomes, such as carbon-credit schemes promoting tree planting in naturally open habitats or subsidies for industrial agriculture, at the expense of grassland biodiversity (Parr et al., 2014; Dinerstein et al., 2019). Furthermore, many African nations, which are the custodians of much of the remaining Pleistocene biodiversity, often lack sufficient climate research funding, limiting their capacity to develop and implement locally appropriate conservation responses. Addressing these structural and regional inequalities is crucial if global policy and action are to meaningfully align climate mitigation efforts with biodiversity conservation. Achieving this alignment requires political willingness and significant economic shifts, informed by an appreciation of Earth's deeper climatic cycles. While reshaping these frameworks is undeniably complex, the magnitude of ecological and ethical stakes demands precisely such transformative action.

## Conclusion and future perspectives

Anthropogenic emissions have amputated the tail of the two-million-year Pleistocene cycle, such that orbital physics will not deliver the next glaciation on any policy-relevant timescale. Conservation, agriculture and climate-mitigation strategies therefore require a dual baseline approach, honouring key ecological functions evolved in the Pleistocene while rigorously screening candidate species and management practices against future climate scenarios. The task would ideally involve:

1. Shielding low-$CO_2$ lineages through assisted persistence or relocation.
2. Managing open biomes explicitly against $CO_2$-driven woody encroachment, even where this strategy challenges conventional carbon sequestration paradigms, by prioritising functional biodiversity and grassland resilience.
3. Screening rewilding/de-extinction candidates for functional resilience *and* climate-fitness; fidelity alone is an insufficient criterion.

Accepting the end of the Pleistocene is intellectually unsettling, but failing to adapt policy to that fact would be biologically catastrophic. Protecting the legacy of 'ice-age survivors' demands coordinated action across scales, from local initiatives to international cooperation, and will need to include novel funding approaches, revised land-use practices and more integrated scientific efforts.

**Open peer review.**   To view the open peer review materials for this article, please visit http://doi.org/10.1017/ext.2025.10006.

**Data availability statement.**   No new data are presented. Figure 1 is plotted from public-domain NOAA data available at http://www.ncei.noaa.gov.

**Acknowledgements.** The authors thank the editors of the de-extinction and rewilding special issue for provoking the writing of this perspective article, and the reviewers for constructive feedback.

**Author contribution.** BWB led the writing, and GFM provided feedback and edits. The original motivations for the Perspective came from a GFM keynote talk on African biodiversity to Stellenbosch University, and work by BWB on Pleistocene and Holocene Australian environments as part of the ARC Centre of Excellence for Australian Biodiversity and Heritage.

**Financial support.** Financial support was provided to B.W.B. from the Australian Research Council (grant DP210101324).

**Competing interests.** BWB is the Editor-in-Chief of *Cambridge Prisms: Extinction*, and GFM is a member of the editorial board. Neither co-author took any role in the editorial process for this submission.

**Ethics.** Not relevant.

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
