## [Reviewer Report]

Review of “Uncooling the Planet: Could Overshooting Paris Accord Targets Erase 3 the Ecological Legacy of the Pleistocene Epoch?”

Line 21 – “Placing the Holocene/Anthropocene debate within a longer Pleistocene context of glacial/inter-glacial cycles frames overshoot more properly within a two-million-year cycle of cool, low-CO₂ conditions.” – The sentence structure is a bit awkward here and I’m unsure how to interpret it. Maybe something more like: “Placing the Holocene/Anthropocene debate within a longer Pleistocene context of glacial/inter-glacial cycles frames the failure to meet Paris Accord targets more properly within a two-million-year cycle of cool, low-CO₂ conditions.”?

Line 25 – “High projected species extinction in biodiversity 26 hotspots and rapid observed ecological changes in subtropical savannas are clear 27 warning flags for low CO2-dependent ecosystems.” – also slightly awkward, consider parsing out. E.g. “rapid observed ecological changes” vs “the rapid pace of observed ecological changes”.

Line 27 – “relevant when” instead of “relevant for”

Line 34 – Could probably be two sentences

Line 78 – This may be a question for the editor, but do we want to use the word “historical” to describe the Pleistocene, or would prehistoric/prehistory be technically correct?

Line 131-140 – The discussion on agriculture only becoming viable under higher CO2 concentrations is interesting but I’m missing some discussion on other (not mutually exclusive) factors driving an increased dependence on agriculture – a big one being the widespread extinction and reduction of large mammals.

Ben-Dor, M. & Barkai, R. (2024) A matter of fat: hunting preferences affected Pleistocene megafaunal extinctions and human evolution. Quaternary Science Reviews 331.

Line 159 – A “few” thousand years is maybe not the right word – the last interglacial lasted perhaps 13 thousand years, with a few generally meaning numbers more in the 3-6 range

Line 243 – This section doesn’t seem to connect well with the preceding part of the manuscript, in my opinion. I know that this paper is being submitted to an issue explicitly about de-extinction, but it kind of reads like the idea is being tacked on to a paper about something else, especially since the Africa focus seems to be abandoned after this point.

I also don’t really buy the overall argument based on the information they’re providing. Rewilding (or at least trophic rewilding as I believe the word is meant here) is explicitly not about recreating past ecosystems, but instead about restoring keystone natural processes. “Pleistocene Park” is an extreme example, but that project has existed since before the word rewilding was even in common usage and has always had more defined goals surrounding things like permafrost and shrubification, both of which are extremely relevant under climate change. What is framed as an “ecological mismatch” is no truer (and perhaps even less true) for megafauna than for smaller species and no more relevant for de-extinction or reintroduction than for traditional conservation. The authors accept that restoration of large herbivores can preserve open habitats, which is true, but then the rest of their argument seems to be based on framing existing megafauna restoration as “nostalgic”, which does nothing to change its effectiveness. We certainly should take climate change into account when discussing megafauna restoration but at the same time, large mammals have been ubiquitous across all non-Antarctic continents for 40+ million years. The authors make the point that we shouldn’t be beholden to a Pleistocene baseline, and I agree, but that is not a prerequisite for megafauna restoration.

Overall, I am afraid that I can’t suggest acceptance of the paper in its current form. This is unfortunate, as I think the premise of the paper was a good one. There are certainly arguments to be made for how de-extinction and rewilding need to take climate change into account or how baselines should be flexible across timeframes and conditions. However, this manuscript seemingly goes out of its way not to make those arguments and is ultimately disorganized, under-cited, lacks concrete examples and thematic consistency, and has an overall feeling of being extremely rushed while also disparaging existing projects without much justification. I know it’s somewhat superfluous to have line edits in a rejection recommendation, but I did enjoy the first sections before the de-extinction segment and was disappointed with the direction the manuscript took after that point. If I am in the minority and the paper goes to revision despite my recommendation, I hope they are considered.

---

## [Reviewer Report]

I agree with the conclusions reached by B.Brook & G.Midgley in their paper and fully share their concerns. I too guessed many years ago that competition between different ecosystems is related to CO2 dynamics. But I can and should add to their reasoning some interesting facts and my own observations from another cold part of the world. In Russia, not only trees but also mosses are expanding because of CO2 growth. Today half of Russia is moss- mosquito and sparse forests and tundra. And during the last glaciation there were rich mammoth steppes and savannahs everywhere. And there are other ways of looking at CO2 growth and warming in Russia. For example, the Russian Academy of Sciences opposed the ratification of both the Kyoto Protocol and the Paris Agreement. Russia is not afraid of warming. Their favorite songs are “oh, frost, frost, don’t frost me...” and “I want summer to never end.”

The issue of climate change is highly politicized, and political change is now more powerful than climate change. President Biden said warming is scarier than nuclear war. And President Trump is withdrawing from the Paris Agreement and plans to invest hundreds of billions of dollars in oil and coal production (CO2 emissions). At the same time, the frozen soils of Russia (including the soils of the mammoth steppes) are the largest reservoir of organic carbon. There is more of it in them than in all coal and oil deposits. No one wanted to believe that the permafrost would soon begin to thaw. But Russia is warming so fast that the permafrost has begun to thaw even in northeastern Siberia - around my home. Seeing how quickly things are changing, I have been publishing mostly in “Rossiyskaya Gazeta” for the past few years. This is the official newspaper of the Russian Federation. I will briefly summarize some facts from these articles.

In the early 70’s there were 20 official climate forecasts, and they all predicted cooling. It was assumed that we were heading towards a new ice age. The revolution happened in 1971 at a symposium of climatologists in Leningrad, when the chief climatologist of the USSR, M.I. Budyko, reported that, according to his calculations, the Earth was beginning to warm because of rising CO2 concentrations. At the same time he published his graphs of future warming and decreasing ice cover of the northern seas. Today we can see that these graphs are remarkably accurate to the actual warming.

Budyko showed that carbon dioxide by itself has little effect on atmospheric temperature. But when the concentration of CO2 changes, a feedback mechanism is activated - because of this, the albedo of the planet changes. And this is the most powerful climatic factor. Changing the albedo of the planet by just 1% changes the global temperature by 20C. Water, forests are dark and absorb up to 90% of the energy coming from the sun. And ice and snow have a very high albedo, they’re white and reflect up to 90% of the energy. More CO2 means a little bit warmer. Because of that, a little less white snow, and because of that more powerful heating, and even less snow....

Western climate scientists met Budyko’s presentation with outrage - this can’t be true. But after that other models showed global warming close in value. And soon it became noticeable and was recognized by all climatologists.

In 1988, there was a severe drought in the United States. Special Senate hearings were devoted to it. James Hansen’s report was particularly noted. It showed that droughts would cover the major grain belts due to rising CO2 and warming. And even in the U.S., farmers will not be able to cope with this drought. The U.S. is in danger of famine. Other Western climatologists supported this prediction, and the Kyoto Protocol was soon adopted. Carbon dioxide became a major threat to civilization.

Forty years ago, there was not only Hansen’s forecast, but also Budyko’s optimistic forecast. The world media at that time, following “New Scientist”, circulated the headline “Soviet climatologist predicts a greenhouse paradise”. According to Budyko’s forecast, the climate at all latitudes would become more comfortable due to CO2 growth, deserts would turn green, and unprecedented harvests would be harvested all over the world. But the USSR collapsed, and the father of global warming and his forecast were forgotten.

More than half a century has passed since Budyko’s landmark report. We have already extracted a third of the available fossil carbon and turned it into CO2. Its content in the atmosphere has increased by 1.5 times. And we can already see which prediction is coming true - Hansen’s or Budyko’s.

Everything is happening as Budyko predicted. On average on the planet since then, the temperature has risen by 10C. Moreover, the closer to the poles, the stronger the increase. In the U.S. temperatures are rising 1.5 times faster than the average for the planet, and in Russia in 2.7 times. At the same time in the very north, in Taimyr, 5 times.

What is a comfortable climate? It is warm and hot in summer, cool and cold in winter. This is familiar. Problems arise when the weather gets out of these habitual frames, when the heat becomes scorching and the frost becomes fierce.

To assess how the comfort level of the climate has changed, one simply has to look at graphs of extreme temperature trends. For the USA, anyone can easily do this, because there is a website www.meteoblue.com, where for many weather stations there is a table, in which for each weather station there is a graph of extreme temperatures.

Fig. 1.

We see that the maximum temperatures both 50 years ago and recently vary by several degrees from year to year, but their values have not increased for half a century! But extreme cold weather everywhere has reliably weakened by a few degrees. All the purple graphs and their linear trends are going up. In the US and around the world, annual average temperatures are rising not because heat waves are increasing, but because frosts are weakening. And average summer temperatures are rising because there are fewer cold days and colder nights in the summer.

Russia is warming almost three times faster than the rest of the world. Huge cold Russia got the lion’s share of global warming. And hot countries, therefore, got a crumb or nothing. For example, New Delhi in India is one of the hottest places on the planet. There’s been no warming here for half a century, neither in winter nor in summer.

The fact that it has warmed a lot in cold regions and frosts have weakened everywhere else is the same phenomenon. Here’s the explanation. The main greenhouse gas on Earth is water vapor. And its content in the air, unlike CO2 and methane, strongly depends on temperature. At 300C in 1m3 of air in the limit, raising the relative humidity to 100%, you can dissolve 30 g of water. But if this air is cooled, vapor will condense - first into water droplets, then into crystals, and at -300C it will remain in the air, despite 100% relative humidity, a hundred times less, only 0,3 g. In a warm atmosphere there is a lot of water vapor, and it, holding the thermal radiation, warms it strongly. Because of this, the vapor content increases even more. But the more vapor there is, the more clouds are likely to appear. And they are white and cool the earth down a lot. Against the background of the struggle between these two powerful opposing factors, the change in CO2 has almost no effect on temperature. And in cold air, there is little vapor, and CO2 becomes the main greenhouse gas. Because of the rise in CO2, the air here is getting a little warmer. Because of this, more water vapor dissolves in it, making the air already noticeably warmer. CO2 is the enemy of frost. It only warms the cold atmosphere noticeably. The climate changes very humanely because of the rise of CO2. It only gets warmer where and when it is welcome.

Winters are getting shorter and milder, and summers are getting longer but not hotter because of it.

Climate disasters are feared by everyone. They’re the norm for our planet. Tree trunks are thick enough to withstand a once-in-a-hundred-year hurricane. And the river valleys are so wide to miss a flood that even the old-timers don’t remember. Climate disasters have and will happen.

The droughts of recent years in the U.S. are presented as an impending catastrophe, a harbinger of the apocalypse. But they are utterly mediocre compared to the drought of a century ago. And in the past, as the rings of thousand-year-old trees tell us, there have been more severe droughts.

The climate is changing, some people are going to have problems because of it. You can’t please everyone. Some need more sunshine, others expect rain. But there is a simple and universal criterion for assessing climate quality.

Our main food is grains. Meat, eggs, milk - they’re all grains, too. Cereals are the basis of our civilization. They’re more demanding of climate than humans. We can hide in the shade, go somewhere for water, plants can’t. Wheat has an optimum temperature for photosynthesis of 20 degrees, rice 27, corn 34. And the limiting temperature for the leaves of most plants, as well as us 420C. If higher, the tissues begin denaturation of proteins. To prevent this from happening, plants have to breathe and sweat intensively, consuming scarce water. Farmers are highly dependent on the weather. Cereals in the fields can burn out, get soaked, or freeze. Strong winds can make grains in the fields fall down, or they can be beaten by hail, or eaten by locusts, or washed away by floods. If grain harvests in the country are falling, it means that the climate is deteriorating and there will be famine. If they are growing, it means that the climate is getting better, more comfortable for crops and therefore for people.

Since 1961, the World Food Organization has been tracking average grain yields in all countries according to a single methodology. On the site www.theglobaleconomy.com all these data are given in the form of convenient graphs. And if you scroll through all of them, you can see that all over the world on all continents the yields have increased almost three times, and it was especially strong in countries with arid climates. The figure shows how grain yields have changed in the fields of the four major grain-producing countries. And it also shows graphs for arid Spain and Algeria. We see a fantastic three- to four-fold increase in yields.

Fig. 2.

What Budyko predicted happened - unprecedented harvests. The planet has warmed by only 1 degree, and such a powerful growth. And there’s no slowdown in sight.

One would think that this growth is the effect of modern “green” technologies. But these technologies have not appeared in all countries, and everywhere has its own specifics of agro-development. For example, the same site shows that in the United States after 1980 they stopped increasing the rate of fertilizers, in Spain - after 2000, and in India and China - continue. But in Algeria, due to armed conflicts after 1985, the rate of fertilizers fell by more than two times. In spite of this, the growth of yields everywhere is equally steady, in parallel with the growth of CO2 concentration in the atmosphere. That means alcohol and bio-diesel could replace oil in the future.

Climate change is bound to cause problems for someone. But the fantastic growth in crop yields overrides them all.

Our planet’s biological carbon cycle is not closed. The carbon content of the atmosphere has decreased 1000 times since it came into existence. Along with sand and silt, dead organics are constantly accumulating in sedimentary strata. Geologic processes are stealing carbon from the biosphere and the atmosphere all the time. Plants absorb CO2 molecules from the atmosphere by their diffusion through open stomata. In doing so, water inevitably evaporates from them. When the first plants came to land, CO2 was plentiful, and plants easily compensated for the loss of water by pumping it through capillaries into their leaves from the soil. Even rare rains were enough to keep arid areas green. At that time, plants competed mainly for light and the one who raised its leaves higher won. And to avoid being eaten, plants were bitter, poisonous, resinous, and prickly.

But the CO2 content of the atmosphere was falling, and to catch one molecule of CO2, plants had to waste hundreds of molecules of water. In many regions, water became the limiting resource for most of the year.

20 million years ago, the biosphere underwent a great revolution with the emergence of grasses and a new type of ecosystem, pasture ecosystems.

Grasses are the youngest plants, the pinnacle of their evolution. They gave up the expensive tall trunk and the waste on thorns, bitterness and poisons. And all resources began to spend on fast-growing leaves and a dense root system, and for this you need a lot of nitrogen, phosphorus and potassium, you need fertile soils. And all this is provided by large herbivores. In their warm, moist stomachs, everything eaten in any climate is quickly converted into fertilizer and returned to the soil. Fast-growing grasses are not afraid of mowing. The more often you mow your lawn, the thicker it becomes.

Even Charles Darwin noted that reproduction in nature many times exceeds the available resources. V.I. Vernadsky formulated this as a law of the pressure of life - everything available must be used. In grasses, even withered, everything is edible for pasture herbivores. What the horses don’t eat, the goats will eat. So everything that has grown over the year must be eaten. And that takes a lot of herbivores. And in wild and domestic pasture ecosystems, in order to mow and turn into fertilizer everything that has grown, the ratio of grass yield to herbivore mass should be 10:1. Therefore, given the yield of grasses, each hectare of pasture must be groomed by centners of herbivores. On the poorest it is one goat and one sheep, and on the richest it is a cow and a horse. On domestic pastures this is true, but in the wild this has only been true in the past.

Vernadsky’s Second Law says that in the course of evolution, those ecosystems that have a higher rate of biocycling win. In an archaic spruce forest, the needles take 10 years to grow, but the trunk takes a hundred. And then this bitter and resinous organic matter, which, except for mold, no one wants to eat, rots on the soil for years. And in a pasture, leaves grow for weeks and decompose in a day.

Grassland ecosystems are the evolutionarily youngest, most highly turnover and most aggressive ecosystems. Grasses feed their defenders. In hungry seasons and bad harvests, elephants, mastodons, and mammoths ate all the withered grass and felled trees and ate the branches. Deer and bulls, eating the bark, killed trees and mowed down their undergrowth. And soon similarly built and outwardly similar pastoral ecosystems, replacing forests, took over the world. Even northern Greenland had rich mastodon savannahs.

The savannahs that emerged because of the decline in CO2 began to influence the climate themselves. To avoid overheating and conserve water, grasses became noticeably lighter in color than trees. When they took over all the continents, the planet became lighter and therefore colder. Because of this, there was more white snow in the north and in the mountains. In the north the forests are dark all year round and the fields are whitish in summer and white in winter, it became even colder.

In the north, the cold soils in the north slowed down the decay of organic matter and began to store carbon. Especially soils under grasses. The world’s forests have 500 Gt of carbon, and the soils (which are mostly soils in the north) have 1500 Gt. In Russian forests carbon in average 5 kg/m2, and in chernozems up to 100. Because of the cold weather, hundreds of gigatons of carbon have accumulated in the soils of the north, and because of this, the CO2 content in the atmosphere has decreased. It got even colder, more white snow. Eventually Greenland became covered with white glacier, the Arctic seas began to freeze and turn white, it got even colder.

(B.Brook & G.Midgley suggest that destroying forests will increase CO2 in the atmosphere and lead to warming. But all models show that in the boreal zone it will lead to severe cooling). Permafrost appeared in the north, and in many places, due to the slow accumulation of dust and silt, the organic-rich soils of the northern savannahs went into permafrost. It took hundreds more gigatons of carbon from the atmosphere. Animals played a big role in this. In pastures with the right density of herbivores, all the snow is overgrazed in winter. The snow loses its insulating properties and the temperature of the soils and permafrost drops by 2-40C because of this. This causes more carbon to accumulate in them. By trimming dark shrubs all year round, herbivores made landscapes lighter. All of this increased the cooling. Mammoth steppes continued to pump carbon from the atmosphere into soils and permafrost.

From the density of bones and skeletons in the permafrost, it was possible to calculate the mass of animals in the pastures of northern Siberia 30-50 thousand years ago. It was colder than today. Nevertheless, here on each square kilometer of pastures on average lived: 5 adult bison, 7 horses, 15 reindeer, 1 mammoth, plus young and rarer sheep, rhinos, antelopes, sheep, lions, wolves, wolverines and even yaks and camels. That’s like today’s richest national parks in Africa, and hundreds of times more than in today’s landscapes of the north.

The stealing of carbon by the subsurface and permafrost 18,000 years ago at the peak of the last glaciation brought atmospheric CO2 concentrations below 0.02%. At that time, thirst and starvation killed most trees on the planet, losing the competition to grains. And even the grains starved, and there were orders of magnitude fewer animals in the north.

This is how Budyko wrote about these dramatic events in his last article published in 2002 (this is his testament article): a little more and the glaciation of the whole planet would have happened, and because of the lack of CO2 all plants would have “suffocated”. The biosphere was close to death. Today, by burning coal and oil, returning what we have stolen, we are rejuvenating the biosphere, returning it to a highly fertile state.

Wild plants, competing with each other, spend most of their extra production on leaf and root growth to intercept light and water from their neighbors. And in agricultural fields, competition is purposely lowered. The amount of moisture entering the soil has not decreased, but the need for it has been reduced. So there is no need to spend a lot of energy and resources on root growth. All these resources can be spent on seed growth. Agrarians, selecting or creating new varieties of cereals with lower costs on roots, achieve a multiple increase in the yield of seeds - grain. CO2 growth is the main reason for the green revolution.

CO2 is the strongest fertilizer. The more CO2, the greener our planet is. Unlike nitrogen, which has to be applied annually, CO2 emission is for centuries. And the Earth is not in danger of overheating due to high concentrations of CO2. It’s never happened. CO2 is a weak greenhouse gas because it does not stop all infrared waves, but only a narrow band of them. But in that band, it’s strong. And it doesn’t take much CO2 to intercept almost all the radiation in it. It’s like covering a window with curtains. The first one traps light strongly, the second one is weaker, and the seventh one has almost no effect. CO2 saturation will happen before all the permafrost melts and loses its carbon.

B.Brook & G.Midgley believe that savannahs appeared because of CO2 reduction and fires. But savannahs are 20 million years old, and coal soot in the deep-sea mud at the bottom of the ocean only appeared half a million years ago. That’s when our ancestors had new technology. In the

past, at the natural, extremely high density of herbivores, in forests and fields, according to the first law of life, everything available was eaten. Everything was mowed, groomed and fertilized, everything was weeded and thinned. Our favorite lawns and park landscapes were everywhere. There was nothing to burn in these ecosystems. With the advent of human hunters, the density of animals decreased, and there was food for fires. Today, there are few animals in nature, and without them, fields and forests are not ecosystems, they are untended, overcrowded plant communities overloaded with dead organic matter. What used to be eaten by herbivores now causes devastating fires.

During the last warming, new hunting technologies emerged and humans spread across the planet, and wherever they appeared, animal populations declined. In North America, 33 species of megafauna disappeared with the advent of man, and in South America, 50 - almost all of them. The biocycle slowed down, soils lost fertility, and in many regions trees, shrubs and mosses replaced defenseless grasses. And on the permafrost of Siberia over the fertile soils of the mammoth steppe a miserable new product was formed - a belt of swampy moss and mosquito sparse forests and tundras.

Grasses are the basis of our civilization. Wheat, corn, rice, sorghum, sugar cane ... - cereals. They’re the main plants in pastures and croplands. They once took over the world, but today there are few of them in the wild. As CO2 rises, there will be even fewer of them. B.Brook & G.Midgley to preserve them, they propose to artificially destroy trees. But grasses have always lived in symbiosis with animals, it’s a grazing ecosystem. And to preserve it, we need to increase the number of animals. For many years we have been conducting experiments to revitalize pasture ecosystems in “the Pleistocene Park”. This is the lower reaches of the Kolyma River in northern Siberia. And in the “Wild Field” park, which is 300 kilometers south of Moscow. And we see that with the right extremely high density of animals, even without mammoths and elephants, they quickly turn forests into savannahs. The same happens in the Oostvaardersplassen park near Amsterdam.

All that needs to be done to revitalize and preserve grassland ecosystems is to give the animals land and will and protect them.

By reviving the richness of nature, we will be able to manage our climate, preserve the biodiversity of key and most important species for our civilization, restore soil fertility and create a food reserve of global importance.

Humanity has many sins against nature. What, then, are we created for? Reasonable man is the only species that can return to the biosphere its main wealth - carbon. Carbon stolen from it by geological processes. And the higher the concentration of CO2, the less deserts there will be and the richer the domestic and wild pasture ecosystems will be.

P.S..

Sick and elderly people are more likely to die when it is very hot or very cold

In July 2021, the Lancet, a major medical journal, published a seminal paper that published the results of global research into this phenomenon. They showed that people on Earth die 10 times more often from cold than from heat, and 25 times more often in hot Africa. This is because humans are genetically a tropical species and are best adapted to heat. We don’t have much fur on us, and we can sweat with our whole body. Our ancestors were daytime hunters and could run

the longest in the heat of the afternoon. Extreme heat due to global warming is not increasing, and frosty and cold days are becoming fewer and fewer. Therefore, global warming is saving millions of lives.

We are shown huge icebergs and scared that the glaciers will melt and everyone will be flooded

Antarctica won’t melt, it’s at the pole. There’s no sunshine for six months. It was covered by a white, self-cooling glacier 35 million years ago. The planet was very warm then. Antarctica is not afraid of the coming warming. On the contrary, the oceans are heating up, evaporation is increasing, there will be more snow in Antarctica and because of this the ocean level may decrease. Icebergs only break off from floating glaciers. All of them may break up quickly, but the sea level (as Archimedes taught) will not change because of this.

Greenland could melt. Because of this, the ocean level will rise by 7 meters. Glaciologists according to different models have calculated that this will take thousands of years. And the average rate of sea level rise will be at most a few mm per year.

Most of the world’s coasts are either rising or falling at a rate of millimeters or even centimeters per year. For example, I live in the mouth of the Kolyma River, it rises 10 mm per year, and the mouth of the Yenisei River, which lies to the west, sinks 20 mm per year. But that doesn’t scare anyone. The melting of Greenland against this background is not a threat.

Mankind itself, by its own will, building dams on the rivers, for the last century very quickly at a rate of many meters per year flooded a huge amount of valuable land and settlements. The problems that our great-grandchildren may have in melting Greenland are negligible against this background.

For glaciers to melt, we need more warm days in the summer. Glaciologists assumed that summers would get warmer, but as it turns out, it’s mostly winters in the north that get warmer. Greenland will melt slower than expected.

To show that I don’t just write articles for the newspaper, here is a list of some of my scientific articles that are used in this review.

1. Zimov, S. A., Chuprynin, V. I., Oreshko, A. P., Chapin III, F. S., Reynolds, J. F., and Chapin, M. C. (1995) Steppe-tundra transition: a herbivore-driven biome shift at the end of the pleistocene. American Naturalist. 146:765-794.

2. Zimov S. A., Zimov N.S., Tikhonov A.N., Chapin F.S. III Mammoth steppe: a high-productivity phenomenon. Quaternary Science Reviews, 2012, V. 57, P. 26-45.

3. Zimov S.A. Pleistocene Park: Return of the Mammoth’s Ecosystem// Science, 2005, Vol. 308. P. 796-798.

4. Sergey A. Zimov, Edward A. G. Schuur, F. Stuart Chapin III. 2006. Permafrost and the Global Carbon Budget. Science, Vol. 312, P.1612-1613.

5. Oleg Anisimov, Sergei Zimov. Thawing permafrost and methane emission in Siberia: Synthesis of observations, reanalysis, and predictive

---

## [Editor Report]

The two reviewers differ on whether the manuscript should be published, although both see value in it. Both reviewers note areas where the manuscript could be improved. Please pay careful attention to the specific critiques of Reviewer 1, particularly the critiques of your framing of rewilding. Reviewer 2 has provided a lengthy commentary. Please take from that review what you feel would help to improve the paper.

---

## [Reviewer Report]

Thank you for the opportunity to review this manuscript. This is an interesting perspective piece—and I hope it will stimulate future thinking about conservation in a non-analog future. I see this has already been through review once before, so I’m happy to report that my comments are all quite minor. Among these (outlined below), one point is worth emphasizing here:

The section entitled “From analogy to prognosis: why functional proxies must pass a future-fitness screen” outlines four criteria that the authors suggest must be considered in light of anticipated future conditions. First, please make it clear that this pertains mainly to terrestrial mammals and not to other organisms. Second, and more importantly, I think these ideas need to be fleshed out and justified in greater detail. It’s one of the more novel contributions of this paper, but the different criteria are only briefly described/justified. If you want people to take these guidelines seriously then more attention is warranted.

Minor edits:

Line 22: “glacial-inter-glacial” should be “glacial-interglacial”

Line 37-38: here you refer to the Holocene as “an unusually prolonged Pleistocene interglacial interval” yet in the introduction (line 63-64) you note that it is “neither warmer nor longer-lived than earlier interglacials” It can’t be both—the latter is a more accurate characterization.

Line 71: interglacial need not be hyphenated as “inter-glacial” (as in lines 62, 63, 70)

Line 73: “anthropogenic impacts” on what?

Line 108-109: I would refer to “succulent Karoo” as an “arid biome” rather than an “arid region”

Line 116-118: but there has been considerable change in African ecosystems through the Pleistocene, especially among large herbivore communities (e.g., https://doi.org/10.1146/annurev-earth-031621-114105)

Line 135: “many plants struggled to achieve meaningful biomass or reliable yields” How well-established is this? Is this not overstated? How could one achieve exceptional megafaunal diversity (relative to today) during glacial Pleistocene if plants are struggling to maintain biomass?

Line 185-186: “orbital forcing would have ended the Holocene in another 10-20 thousand years.” That would make the Holocene 22-32 ka long—which would be exceptional for an interglacial.

Line 210-212: yes but those organisms also endured numerous exceptionally rapid climate shifts (e.g., Dansgaard-Oeschger events)

Line 222: there is an extra period here.

Line 243: isn’t this reference not this a case study of Asian ecosystems?

Line 272-296: I would love to see the ideas outlined in this section fleshed out in more detail. If the goal is to encourage conservationists and land managers to adopt these guidelines, then it is probably necessary. [see above]

Line 279: can you more clearly outline what you mean by “proxy species?”

Line 340: a space is needed before the reference

Line 350-352: but what about fire? How does that fit into this?

Line 514: fix typo in title

---

## [Editor Report]

We have received one final review of the manuscript, which includes a couple more issues and minor edits. Once you’ve addressed these, the paper should be ready for publication.

---

## [Editor Report]

Thank you for addressing the recommended changes. I am happy to accept your paper for publication.